# Shoreface erosion counters blue carbon accumulation in transgressive barrier-island systems

Mary Bryan Barksdale [1] ✉, Christopher J. Hein [1] & Matthew L. Kirwan [1]

Landward migration of coastal ecosystems in response to sea-level rise is altering coastal carbon dynamics. Although such landscapes rapidly accumulate soil carbon, barrier-island migration jeopardizes long-term storage through burial and exposure of organic-rich backbarrier deposits along the lower beach and shoreface. Here, we quantify the carbon flux associated with the seaside erosion of backbarrier lagoon and peat deposits along the Virginia Atlantic Coast. Barrier transgression leads to the release of approximately 26.1 Gg of organic carbon annually. Recent (1994–2017 C.E.) erosion rates exceed annual soil carbon accumulation rates (1984–2020) in adjacent backbarrier ecosystems by approximately 30%. Additionally, shoreface erosion of thick lagoon sediments accounts for >80% of total carbon losses, despite containing lower carbon densities than overlying salt marsh peat. Together, these results emphasize the impermanence of carbon stored in coastal environments and suggest that existing landscape-scale carbon budgets may overstate the magnitude of the coastal carbon sink.

The coastal landscape is widely recognized for its ability to store organic matter in blue carbon ecosystems, such as salt marshes and seagrass beds, that bury carbon (C) in soils and sediments at rates orders of magnitude greater than terrestrial systems[1]. Sea-level rise (SLR) is thought to augment the coastal C sink[2], especially in marshes that are building soils vertically at rates similar to those of relative SLR[3–5]. A direct coupling between SLR and soil C accumulation can result in increases in C stocks even where marshes are eroding[2,6]. However, the capacity of the coastal zone to store blue carbon over centuries to millennia under rapid rates of SLR remains uncertain. For example, rapid SLR can exacerbate inundation stress and eventually lead to drowning of intertidal blue carbon coastal ecosystems, thereby reducing sequestration potential while also degrading soil C[7–9]. Additionally, SLR can lead to large C losses within the coastal zone by driving ecosystem transgression (for example, forest retreat, which prompts substantial aboveground biomass loss[10,11]) and/or by driving erosion of C-rich sediments when exposed along open-ocean coasts[12,13]. Thus, coastal landscapes facing the combined threats of SLR and erosion risk a blue carbon stock that is both diminished and more fleeting.

Barrier-island beach and dune systems protect the C-rich sediments of backbarrier marsh from wave erosion along many coasts globally and can supply sediments to fringing backbarrier marsh during high-energy events[14–16], processes that support lateral and vertical resilience to SLR, respectively. However, this supportive function of barrier islands is jeopardized by SLR, which, compounded with intensifying coastal storms and sediment deprivation, forces oceanside barrier shorelines to transgress (through island narrowing via erosion and/or wholesale landward migration) at accelerating rates[17,18]. Soil C stocks previously protected by barrier islands are eventually exposed and subjected to high-energy, open-ocean processes, possibly shifting transgressive barrier-island systems from C sinks to C sources[12].

Across the coastal landscape, the magnitude of the net C sink depends on the balance[19] between C loss due to erosion or drowning, and C accumulation in ecosystems migrating and/or accreting apace with SLR[2,6,11,20]. However, these landscape-scale C budgets typically focus on the evolution only of vegetated ecosystems, and assume shallow depths of erosion, as is common in protected environments. In contrast, wave action along open-ocean shorefaces can rework

[1]Virginia Institute of Marine Science, William & Mary, P.O. Box 1346 Gloucester Point, VA 23062, USA. ✉e-mail: mbarksdale@vims.edu

sediments well below mean sea level, exposing to erosion not only surficial salt marsh peat, but also far deeper sedimentary deposits. Failure to account for these processes may lead to large overestimates of C storage in coastal ecosystems.

Here, we combine geospatial data of barrier island retreat rates, organic carbon (OC) accumulation rates within backbarrier marsh soils and seagrass and lagoon sediments, and the OC content of eroding sedimentary facies to develop a regional-scale OC budget for the rapidly transgressing Virginia Atlantic coast (USA). Sedimentologic and geochemical analyses of 10 new sediment cores (each 3–19 m long) together with additional published stratigraphic data were used to determine facies-specific thicknesses, OC densities, and OC erosion rates (Fig. 1; eq. [1]). We find that buried lagoon sediments associated with unvegetated environments contribute the vast majority (>80%) of OC eroded on the beach and shoreface of transgressing barrier islands. Moreover, we find that erosion of these deep deposits leads to rates of OC loss that exceed annual OC accumulation summed across the entire backbarrier environment, despite the well-known capacity of blue carbon ecosystems to sequester OC.

## Results and discussion
### Barrier island stratigraphy and carbon characteristics
The largely undeveloped and rapidly transgressing Virginia Barrier Islands (VBI) are located in the mid-Atlantic SLR hotspot[21] and generally characterized by either wholesale landward migration or rotation of formerly progradational islands[22] (Fig. 1a). Stratigraphic and OC

analyses reveal that those islands which are migrating landward are characterized by thin (<2 m thick) sandy beach and dune deposits[22] perched atop discontinuous, thin (~0.9 m) marsh peat and thick (~6.6 m) lagoon deposits (Fig. 1c). In contrast, former backbarrier peats associated with historically progradational islands (Parramore, Hog) were long-ago eroded as those islands migrated to their landward-most positions, leaving only thinner (0.75–6.25 m) remnant lagoon deposits preserved under relatively thick (~4.5 m) barrier sands[22]. Averaged across the seven migrating islands, the beachface-exposed marsh is 0.9 m thick (ranging from 0.6 [Smith] to 1.3 m [Assawoman]) and characterized by a relatively homogenous mixture of marsh roots and silt- or clay-dominant minerogenic sediment with an average OC density of 26.8 kg OC m$^{-3}$ (ranging from 23.3 [Smith] to 31.5 kg OC m$^{-3}$ [Cobb]; Fig. 1c; Table 1). In contrast, lagoon deposits consist of a complex set of facies ranging from clay to medium sand, predominantly very dark greenish grey in color, with frequent shell fragments. Across all ten islands, the average lagoon deposit thickness is 6.0 m (varying between 3.5 [Parramore] to 8.5 m [Wreck]), and the average lagoon OC density is 7.6 kg OC m$^{-3}$ (ranging from 5.3 [Smith] to 10.1 kg OC m$^{-3}$ [Cobb]; Fig. 1c). Sandy units interbedded within lagoon complexes average 0.8 m of very fine to very coarse sand (ranging from 0.0 [Assawoman, Cobb, Myrtle] to 1.6 m [Metompkin and Cedar]). We estimate that 38.8 km$^2$ of backbarrier marsh was buried and re-exposed by island migration along the island chain from northern Assawoman to southern Smith between 1870 and 2017 C.E., at a system-wide rate averaging 0.26 km$^2$ per year.

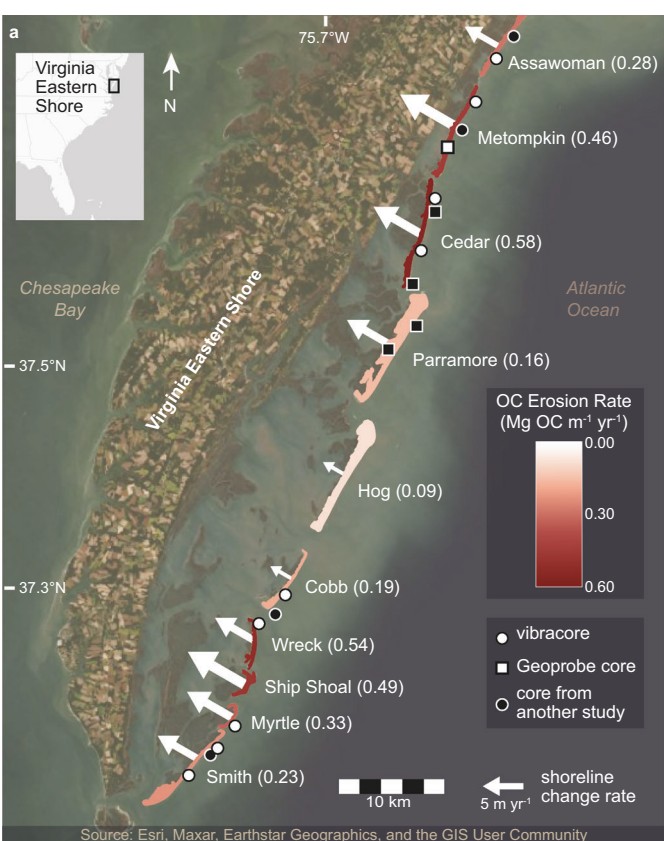

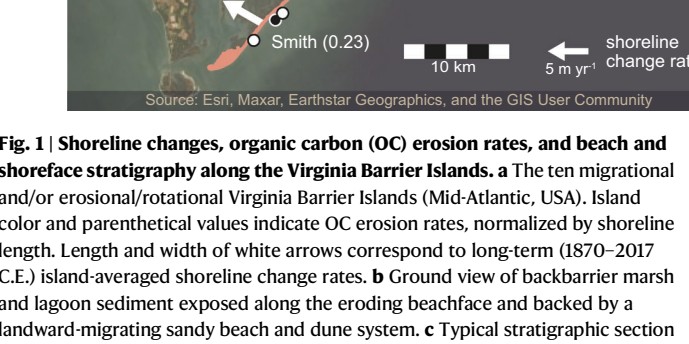

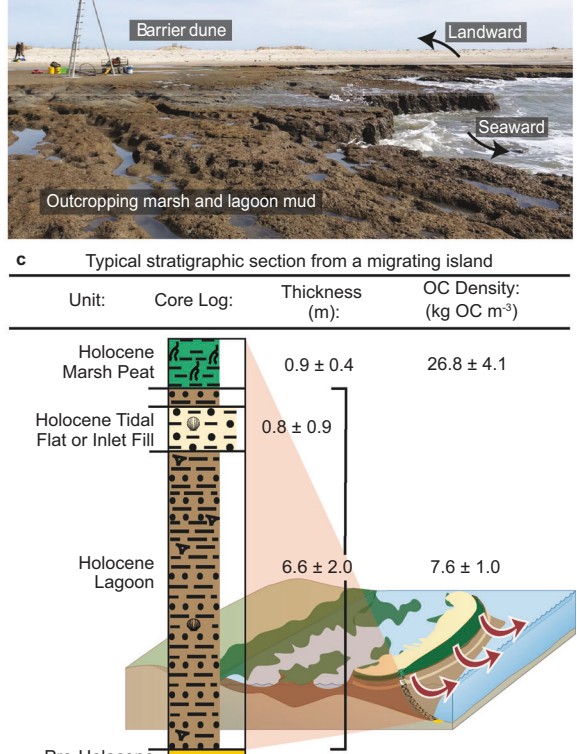

**Fig. 1 | Shoreline changes, organic carbon (OC) erosion rates, and beach and shoreface stratigraphy along the Virginia Barrier Islands. a** The ten migrational and/or erosional/rotational Virginia Barrier Islands (Mid-Atlantic, USA). Island color and parenthetical values indicate OC erosion rates, normalized by shoreline length. Length and width of white arrows correspond to long-term (1870–2017 C.E.) island-averaged shoreline change rates. **b** Ground view of backbarrier marsh and lagoon sediment exposed along the eroding beachface and backed by a landward-migrating sandy beach and dune system. **c** Typical stratigraphic section

from sediment cores penetrating through beachface-exposed marsh (as in **b**) along a landward-migrating island, identifying stratigraphic units with associated average thicknesses (with standard errors) and OC densities (with uncertainties that account for propagations of sediment bulk density standard errors and 95% confidence intervals of organic matter to OC conversions; see Supplementary Information). Barrier system diagram modified from Tracey Saxby, Integration and Application Network (ian.umces.edu/media-library).

**Table 1 | Variables Used to Calculate Long-Term (1870–2017) Organic Carbon (OC) Erosion Rates for the Virginia Barrier Islands**

| Island | Marsh | | | Lagoon | | | | Combined |
|---|---|---|---|---|---|---|---|---|
| | Thickness (m) | Exposure rate (m² yr⁻¹) | OC density (kg OC m⁻³) | Thickness (m) | Shoreline change rate (m yr⁻¹) | 1870 island length (m) | OC density (kg OC m⁻³) | OC erosion rate (Gg OC yr⁻¹) |
| Assa. | 1.26 ± 0.25 | 14243 ± 672 | 23.6 ± 3.3 | 5.63 ± 0.38 | 4.74 ± 0.68 | 6599 ± 16 | 8.2 ± 1.0 | 1.86 ± 0.31 |
| Met. | 0.66 ± 0.16 | 53135 ± 2508 | 26.8 ± 3.9 | 7.26 ± 3.06 | 7.67 ± 1.45 | 11442 ± 16 | 6.7 ± 0.8 | 5.21 ± 2.06 |
| Cedar | 1.07 ± 0.27 | 41734 ± 1970 | 27.6 ± 4.0 | 7.63 ± 3.38 | 6.68 ± 1.37 | 10687 ± 16 | 9.1 ± 1.2 | 6.18 ± 2.52 |
| Parra. | 0.90* ± 0.36* | 161 ± 8 | 26.8* ± 4.1* | 3.50 ± 2.75 | 5.94 ± 1.82 | 13000 ± 16 | 7.6* ± 1.0* | 2.04 ± 1.74 |
| Hog | 0.90* ± 0.36* | 3775 ± 178 | 26.8* ± 4.1* | 3.50‡ ± 2.75‡ | 3.10 ± 1.95 | 11288 ± 16 | 7.6* ± 1.0* | 1.01 ± 0.94 |
| Cobb | 0.98 ± 0.36* | 15909 ± 751 | 31.5 ± 4.4 | 4.55 ± 1.98* | 3.03 ± 2.77 | 10256 ± 16 | 10.1 ± 1.3 | 1.92 ± 1.47 |
| Wreck | 1.18 ± 0.88 | 22004 ± 1039 | 27.1 ± 3.9 | 8.47 ± 2.82 | 5.72 ± 3.21 | 3934 ± 16 | 7.5 ± 0.9 | 2.13 ± 1.09 |
| S.S. | 0.90* ± 0.36* | 12432 ± 587 | 26.8* ± 4.1* | 6.63* ± 1.98* | 8.07 ± 3.14 | 3405 ± 16 | 7.6* ± 1.0* | 1.68 ± 0.71 |
| Myrtle | 0.60 ± 0.36* | 15240 ± 719 | 27.5 ± 4.3 | 6.90 ± 1.98* | 6.38 ± 2.35 | 3659 ± 16 | 5.9 ± 0.8 | 1.20 ± 0.49 |
| Smith | 0.56 ± 0.24 | 46484 ± 2194 | 23.3 ± 4.8 | 5.98‡ ± 0.29 | 5.63 ± 1.01 | 12633 ± 16 | 5.3 ± 0.7 | 2.87 ± 0.58 |
| | | | | | | | Combined Virginia Barrier Islands = 26.12 ± 4.36 | |

*Assa.* Assawoman, *Met.* Metompkin, *Parra.* Parramore, *S.S.* Ship Shoal. For more information on how uncertainties and standards of error were calculated, refer to Supplementary Information.

*Based on the average of all migrating Virginia Barrier Islands due to a lack of cores or due to a lack of multiple island-specific cores when calculating uncertainty values.

‡Based on Parramore averages.

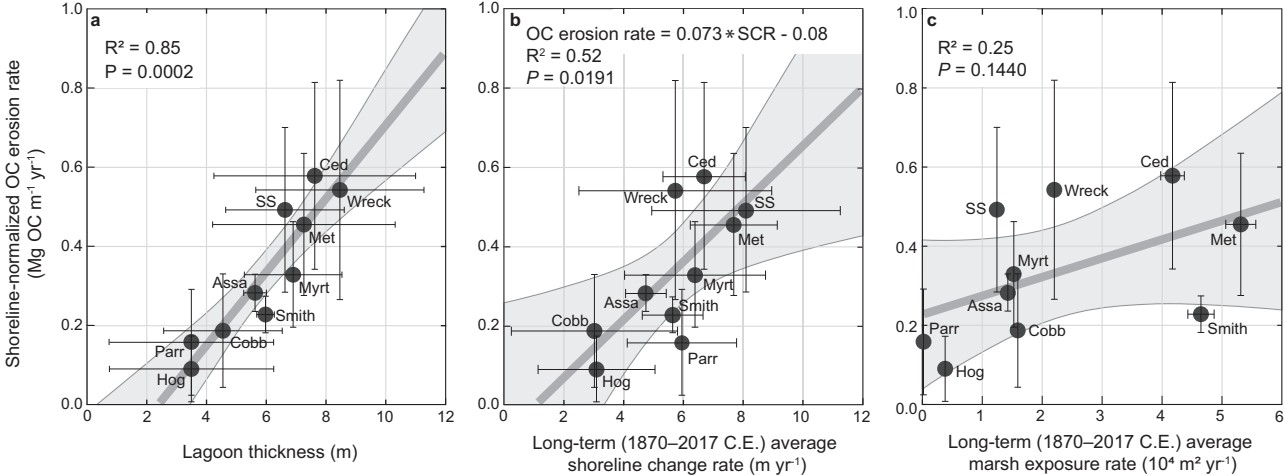

**Fig. 2 | Drivers of organic carbon (OC) erosion rates along the beach and shorefaces of the Virginia Barrier Islands.** Shown are regressions between shoreline-normalized OC erosion rates and: (**a**) Lagoon thickness with horizontal error bars representing standard error; (**b**) Long-term (1870–2017 C.E.) shoreline change rates (SCR) with horizontal error bars representing the average of all transect 90% confidence intervals for each island (see Supplementary Information); and (**c**) Long-term average marsh exposure rates with horizontal error bars representing total uncertainty of marsh exposure rates (see Supplementary Information). Error bars smaller than symbols are not shown. Vertical error bars represent uncertainty propagations for Eq. (1) outputs. Solid lines indicate fitted linear regressions; gray windows demarcate 95% confidence intervals. Island abbreviations: Assa Assawoman; Met Metompkin; Ced Cedar; Parr Parramore; SS Ship Shoal; Myrt Myrtle.

Although marsh peat is widely recognized for its large blue carbon stores[1], we find that lagoon facies thickness is the single largest driver of shoreline-normalized OC erosion rates (equation [1]), accounting for 85% of variability ($P < 0.001$; Fig. 2a). Shoreline-change rate accounts for approximately half of the variability in OC erosion rates ($R^2 = 0.52$; $P = 0.02$; Fig. 2b). In contrast, neither the rate of long-term average marsh exposure ($P = 0.14$; Fig. 2c) nor marsh or lagoon OC densities (Supplementary Fig. 1) have a significant effect on OC erosion rates.

Applying new multi-decadal and island-specific shoreline-change rates, marsh-exposure rates, and island shoreline lengths to Eq. (1) (Supplementary Tables 1–3), we find that beach/shoreface OC erosion has accelerated over shorter time periods (Fig. 3), reaching an annual average rate of $42.9 \pm 10.0$ Gg OC yr⁻¹ between 1994 and 2017 C.E. This is more than 125% greater than the average annual OC accumulation for the entire VBI backbarrier—including OC accumulated in marsh,

seagrass, and lagoon soil/sediment—over a similar time period ($33.8 \pm 6.0$ Gg OC yr⁻¹; 1984–2020 C.E.)[23–25] (Fig. 3; Supplementary Table 4).

## Implications for coastal carbon budgets

Carbon budgets that cross traditional ecosystem boundaries are crucial for establishing the degree to which coastal landscapes can mitigate climate change through C sequestration[11]. Recent studies demonstrate that ecosystem transitions associated with SLR (for example, conversions of forest to marsh or of marsh to open water) lead to shifts in magnitudes and loci of C burial and C loss[2,6,11,20]. However, such landscape-scale C budgets typically focus on vegetated ecosystems and include C loss due to marsh submergence or erosion only to a depth of 1 m[20,26,27]. Thus, widely-used protocols for assessing vulnerability of C stocks often overlook sediment C accumulation in unvegetated systems as well as C loss due to deeper erosion of non-

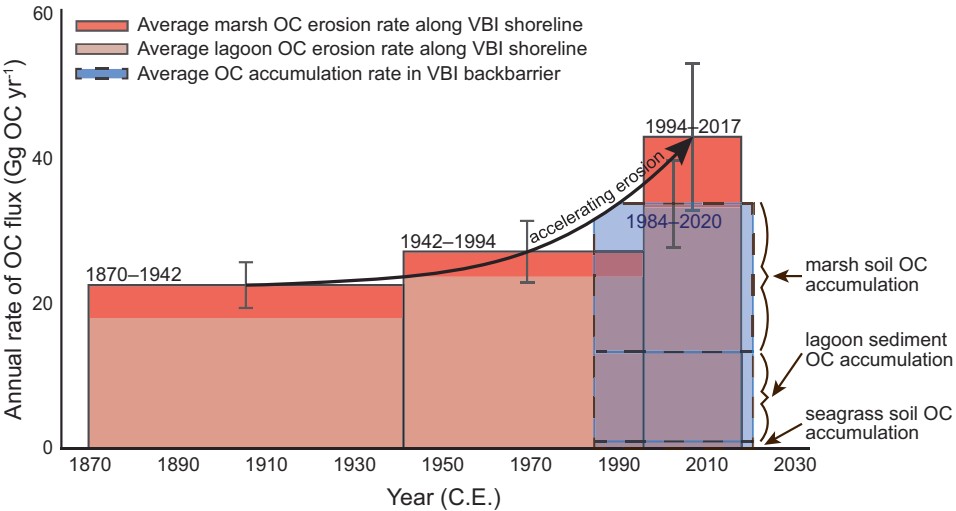

**Fig. 3 | Multi-decadal organic carbon (OC) erosion and accumulation rates for the Virginia Barrier Islands.** Rates of annual OC flux in the Virginia Barrier Islands (VBI) between 1870 and 2017 C.E. Gray bars for OC erosion rates represent uncertainty propagations for the sum of all island-specific Eq. (1) outputs over each time period. The gray bar for the OC accumulation rate represents uncertainty propagations associated with mapping and soil OC measurements, following ref. 23 (see Supplementary Information).

vegetated facies. Here, by extending the landscape C budget to include sites of sediment/soil OC accumulation and erosion that traditionally have been ignored, we find that backbarrier lagoon and tidal-flat sediments contribute >80% of the total annual OC eroded in the VBI system (Fig. 3). Thus, incorporating these sediments into OC flux estimates not only magnifies the OC erosion term in our budget but also challenges previous understandings of the role deep, unvegetated sediments play in the coastal OC sink.

Organic C capture in vegetated ecosystems has been the paradigm of coastal OC research since the term 'blue carbon' was first coined in the early 2000s[1,7,28]. However, emerging evidence demonstrates that non-vegetated and subtidal coastal environments can contain substantial OC stocks[29,30], fed by the deposition of particulate matter (for example, organic matter from nearby erosion of vegetated systems or from productivity within the overlying water column) and in situ microphytobenthic productivity[31,32], as has been shown for the VBI lagoons[33]. We find that, despite hosting OC densities that are approximately one-third of that of the marsh (Fig. 1c), the thickness of lagoon deposits is a more important driver of OC erosion fluxes than factors that commonly garner more attention, such as marsh OC density, marsh thickness, or marsh erosion rate (Fig. 2a and c; Supplementary Fig. 1). This aligns with emerging evidence that unvegetated coastal areas are important components of the coastal OC budget, and can, depending on their areal extent and thickness, account for more total OC storage than vegetated areas within the same landscape. In fact, we find that just the average annual erosion of lagoon OC ($33.4 \pm 9.8$ Gg OC yr$^{-1}$; 1994–2017 C.E.) could negate the OC accumulated annually in the entire backbarrier averaged over a similar time period ($33.8 \pm 6.0$ Gg OC yr$^{-1}$; 1984–2020 C.E.) (Fig. 3).

The disproportionately high rates of OC burial in coastal ecosystems[1] leave large pools of OC subject to destabilization following rapid SLR and commensurate wetland drowning, forest dieback, and/or enhanced erosion[2,7,10]. Previous work by ref. 12 considered an additional consequence of SLR on OC storage (that is, transgression of barrier islands) and found that erosion of outcropping salt marsh along barrier-island beach and shorefaces can flip the system from a C sink to a C source. Likewise, our quantification of the most recent (1994–2017 C.E.) rate of annual OC erosion along the VBI shoreface is approximately 1.3 times the rate of OC accumulation across the entire VBI backbarrier over a similar time

period[23–25] (Fig. 3; Supplementary Table 4). Including only marsh soil OC in these budgets would erroneously suggest that the VBI remains a strong sink for OC, netting an average 11.5 Gg OC yr$^{-1}$ over the past two decades (Fig. 3; Supplementary Information). Like other landscape-scale carbon budgets[6,12,20,23], our work assumes that eroded carbon represents a source of carbon to the atmosphere or to non-coastal ecosystems. However, fully classifying the VBI chain as a net OC source would require tracking the fate of this shoreface-eroded OC, which may include remineralization, offshore burial, or possibly transport and redistribution to the backbarrier through tidal inlets. Nevertheless, the imbalance we measure between annual rates of backbarrier OC accumulation and shoreface OC erosion implies that, at the very least, barrier-island transgression results in a coastal OC sink that is far more tenuous than commonly assumed.

**Feedbacks between blue carbon and climate.** Blue C storage dynamics have traditionally been considered a negative climate feedback, whereby SLR drives enhanced soil OC accumulation in coastal ecosystems like salt marshes[2–4,6,9,34]. For the VBI, we find that an increase in the rate of island transgression by only 1 m yr$^{-1}$ intensifies OC erosion by approximately 73 kg OC m$^{-1}$ yr$^{-1}$ (Fig. 2b). Thus, our results confuscate the current understanding of coastal OC processes by suggesting that dynamics along open-ocean coasts can constitute a positive climate feedback. Given newly uncovered multi-decadal lags in barrier response to SLR[18], our findings suggest that OC erosion along migrating barrier islands will continue to accelerate as island movement equilibrates to modern (and even faster, future) rates of SLR. Narrowly focusing on OC gains and losses within the top meter of vegetated environments underestimates the OC potentially eroded from deeper and unvegetated ecosystems, especially within dynamic coastal systems. Therefore, landscape-scale OC budgets based on the evolution of shallow, vegetated environments may obscure the potential for coastal landscapes to switch from net C sinks to C sources, a threshold which the VBI may already have crossed. Regardless of the magnitudes and sites of OC accumulation and erosion, our findings demonstrate that, for systems in which barrier islands are free to move landward, blue carbon stored in wetland and thick lagoon sediments is largely ephemeral.

## Methods

### Shoreline behavior

The Virginia Barrier Islands (VBI) comprise a 110-km-long chain of 12 mixed-energy islands backed by salt marsh and shallow lagoons along the US Mid-Atlantic Coast (Fig. 1a). The absence of artificial shoreline stabilization along all but Wallops Island allows most to erode and/or migrate landward in response to storms and SLR, which they do at an average rate of 4.35 m yr$^{-1}$ (1851–2017)[18]. Excluding net-progradational Fisherman's Island (located at the southern longshore depocenter at the mouth of Chesapeake Bay), individual island shorelines transgress at rates between 3.1 m yr$^{-1}$ (Cobb) and 7.5 m yr$^{-1}$ (Ship Shoal)[18] (Fig. 1a). This process exposes expansive marsh deposits along the seaward side of many of these islands (Fig. 1b) and, visible at very low tide, lagoon deposits along the marsh periphery or directly under barrier sands.

### Sediment core analyses

Nine vibracores (each 3–9 m long) and one GeoProbe core (19 m long) collected from across seven islands (Fig. 1a) were analyzed for organic-matter (OM) content via loss-on-ignition (LOI) and grain size, and a subset for total organic carbon (TOC) content (Supplementary Information). We apply the resulting marsh- and lagoon-specific conversion factors (Supplementary Fig. 2) to approximate OC content based on OM values for all downcore samples.

### OC erosion rate calculations

Contact between the marsh and lagoon unit, as well as the base of the Holocene barrier-system were determined according to sediment texture, mineralogy, and OM content, in keeping with the unit descriptions of refs. 35,36. We estimated OC erosion rates (g OC yr$^{-1}$) associated with loss of both marsh and lagoon deposits for each island as:

$$OC\ erosion\ rate = \left(T_{marsh} * ER_{marsh} * \rho OC_{marsh}\right) \\ + \left(T_{lagoon} * L_{shoreline} * SCR * \rho OC_{lagoon}\right) \quad (1)$$

where, following ref. 12, we apply island-average OC densities, $\rho_{OCx}$ (g OC m$^{-3}$), to the island-average thicknesses, $T_x$ (m), of the marsh and lagoon units based on new and published cores[35,37–40] (Fig. 1c; Table 1; Supplementary Table 5). Unlike ref. 12, however, we account for lagoon sediment OC in our erosion terms, quantifying a maximum blue carbon loss term for erosion of the entire Holocene unit. Except where replaced by inlet fills, lagoon deposits ubiquitously underlie both transgressive and progradational islands within the VBI chain[35,36,39,41]. Thus, lagoon sediment volume loss is approximated by multiplying the shoreline length, $L_{shoreline}$ (m) (Supplementary Table 1), by the island-specific shoreline-change rate, $SCR$ (m yr$^{-1}$) (Supplementary Table 2). In contrast, beach/shoreface marsh erosion is confined to discontinuous portions of migrating islands. Following ref. 42, we used the earliest-mapped backbarrier marsh extent and overlaid successive island positions up to 2017 C.E. to calculate a time-averaged annual marsh exposure rate due to island transgression, $ER_{marsh}$ (m$^2$ yr$^{-1}$) (Supplementary Table 3). We used Digital Shoreline Analysis System (DSAS)[43] to calculate shoreline positions at 50-m spaced transects along the length of the VBI to calculate both long-term (1870–2017 C.E.) and short-term (1870–1942; 1942–1994; 1994–2017) shoreline-change rates, $SCR$. System-wide rates are valued as the sum of component islands.

## Data availability

The short-term OC erosion rates, OM-to-TOC conversions, sediment core descriptions, and sediment core OC calculation data generated in this study have been deposited in the EDI Data Repository (https://doi.org/10.6073/pasta/547b7f5ba77fd99172a5564f8beb7b62)[44].

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

## Acknowledgements

This work was supported by National Science Foundation Geomorphology and Land Use Dynamics Program (EAR-2022987) (CJH), and the National Science Foundation CAREER, LTER, and CZN programs (EAR-1654374, DEB-1832221, and EAR-2012670) (MLK). We thank S. Fate, J. Lewis, C. Clarke, J. Garber, E. Hein, K. Holcomb, N. Ingle, T. Meredith, T. Messerschmidt, G. Molino, and C. Obara for assistance collecting cores as well as J. Connell, K. Cahoon, A. Gravgaard, and G. Weeks for assistance in core collection and processing. We thank C. Barksdale and M. Barksdale for manuscript edits as well as M. LaGanke for manuscript edits and assistance in core processing. Access to all sites was granted through partnership with The Nature Conservancy and U.S. Fish and Wildlife Service. In particular, we thank K. Holcomb, A. Wilke, S. Bates, and S. Miller for assistance accessing field sites and would like to thank all past and current stewards of the VBI lands and waters.

## Author contributions

C.J.H. and M.L.K. conceptualized the study. M.B.B. and C.J.H. conducted fieldwork. M.B.B. conducted all labwork, analyzed data, and wrote initial manuscript. All three interpreted results and contributed to manuscript writing and editing.

## Competing interests

The authors declare no competing interests.
