## [Peer Review File · Nature Communications]

Shoreface erosion counters blue carbon accumulation in transgressive barrier-island systemsREVIEWER COMMENTS

Reviewer #1 (Remarks to the Author):

This manuscript calculates the loss of OC associated with erosion of the unvegetated environments within transgressing barrier islands. The manuscript characterizes a previously unaccounted for flux of OC from barrier islands into the coastal ocean associated with accelerating erosion and emphasizes further scrutiny of the role deep, unvegetated sediments play in coastal carbon dynamics. The provided supplementary materials and methods substantiate the conclusions and claims of the paper. I would appreciate a section specifically outlining how organic carbon was determined on the subset of sediments used to construct the conversion equation relating OC to LOI (see below). Though the manuscript relies heavily both on previously collected data and on a series of assumptions, the reasoning for each is clearly outlined and uncertainty is openly described throughout. After careful scrutiny of the paper, I find it well-written and informative with no major flaws that would preclude publication. The manuscript is well-reasoned with a sound methodology and is a valuable addition to the blue carbon and barrier island erosion literature. Beyond that, it is an entertaining and insightful manuscript and I commend the authors on an excellent intellectual effort.

I have a few minor comments that are outlined below:

Lines 41 – 49. I understand why you're distinguishing between barrier islands and backbarrier marsh, but the two are not mutually exclusive. The backbarrier marsh is a component of the barrier island, which makes this paragraph a bit confusing. I'd suggest modifying the terminology to be more specific when you mention 'barrier island' to alleviate any confusion on the part of the reader.

How was total organic carbon evaluated? I don't see any material about how these sediment subsamples were analyzed for TOC. Was inorganic carbon removed with acid? There seem to be quite a few points that seem anomalously high in TOC %, relative to LOI %. Could this be residual carbonate?

Supplemental Data, Line 114: Can you define 'surface marsh'? What is the depth on this?

Supplemental Data, Table 4: I'm a bit confused as to why the cores for Cedar Marsh are sporadically used. Why use this data for some things and not others?

Supplemental Data, Table 5: What are the notations (asterisks, other characters) for in the CAR column?

While I understand that this is likely a mechanism of a slower publishing process, it is tough to evaluate the use of carbon accumulation rates (quite heavily) based on a study that is currently still in review at another journal and thus inaccessible. As a result, I can't speak to the appropriateness of using these measurements.

Throughout the manuscript, there are instances where "OC" and "C" appear to be used interchangeably. For example:

Line 194: "however, we account for lagoon sediment OC in our erosion terms, quantifying a maximum blue C loss term for erosion of the entire Holocene unit".

Line 164: "blue C stored in wetland and thick lagoon sediments is largely ephemeral".

What about inorganic carbon?

Reviewer #2 (Remarks to the Author):

This paper describes a study of estimating sediment/soil organic carbon erosion fluxes of the Virginia Barrier Islands that have habitats of back barrier marshes, subtidal lagoon sediments and subtidal sediments with sea grasses. The researchers took sediment cores to estimate carbon content and how the erosion of these sediments and soils can help assess the annual rate of eroded organic carbon flux in this system. The main result in the study shows that the lagoon carbon erosion rate is higher than the soil/sediment carbon accumulation rate and thus suggesting that erosional processes can reduce the magnitude of net carbon sinks in these coastal landscapes. The authors argue that many studies focus on vegetated habitats but that these other subtidal sediment habitats are important to assess as carbon is likely eroding from them and could be influencing the net carbon

budgets. This is a novel study because it includes non-vegetated habitats, and the methods are well described, and the results and discussion are well written. I provide many suggestions (by line numbers) to help the authors clearly explain their findings.

Manuscript - suggestions with Line numbers:

11 – change “strength” to “magnitude”

12: - add “soil” before “carbon” so it reads, “soil carbon”

18: - add “soil” before “carbon” so it reads, “soil carbon”

22: add “soil and sediment carbon” , so it reads “the impermanence of soil and sediment carbon” – I think this is important to remind the reader of where the buried or stored carbon is located.

24: Change “organic-rich sediment” to “organic-rich marsh soils and lagoon sediments” – to emphasize the 2 types of buried carbon this study evaluated.

24: I also think “magnitude” is a better term than “strength” listed here.

28: add “sediments and soils” after “bury C” so it reads, “bury C in sediments and soils” at rates...

29: Note you use “magnitude” here and I think this is the best term for your argument and study and I would suggest to only use this term in this paper and not “strength” to keep consistent and easy to read.

31: add “soil” so it reads “soil C accumulation can result”

40: I don’t know what these terms “diminished and more fleeting” mean scientifically, I would simplify this and state “risk a blue C stock that is decreasing in time and space”

50: add “net” so it reads “magnitude of the net C sink”...

50-51: The magnitude of the net C sink can also depend on other fluxes like the net methane flux. I would add the Chapin et al. 2006 citation to this sentence. Also note that the flux of net lateral transfer or flux of particulate carbon is a term commonly used in these carbon budgets and this term includes erosion. Thus, this study helps to estimate that flux of particulate carbon.

Chapin, F. S., G. M. Woodwell, J. T. Randerson, E. B. Rastetter, G. M. Lovett, D. D. Baldocchi, D. A. Clark, M. E. Harmon, D. S. Schimel, R. Valentini, C. Wirth, J. D. Aber, J. J. Cole, M. L. Goulden, J. W. Harden, M. Heimann, R. W. Howarth, P. A. Matson, A. D. McGuire, J. M. Melillo, H. A. Mooney, J. C. Neff, R. A. Houghton, M. L. Pace, M. G. Ryan, S. W. Running, O. E.

Sala, W. H. Schlesinger, and E.-D. Schulze. 2006. Reconciling carbon-cycle concepts, terminology, and methods. *Ecosystems* 9:1041–1050.

And you can add that citation (by a superscript) to the word “balance” as this paper describes fluxes to measure for estimating the net ecosystem carbon balance.

53: Agree with this statement that carbon budgets typically focus on the vegetated ecosystems.

58: use the term “backbarrier marsh C soil accumulation rates” – marsh and soil need to be added here to be super clear to the reader.

63: Equation 1 was difficult to find, I suggest adding “equation” in front of the “(1)” on page 8 or line number 191.

96: edit “marsh exposure” to “long-term average marsh exposure rate” to be consistent with your figure.

97: Can you cite the figure or table that backs up this statement?

101: In Figure 3 it lists 1994 and not 1997. So should this be 1994?

102: add “soil/sediment” so it reads “annual soil/sediment OC accumulation rate”

102: It lists 33.8 and on Figure 3 it says 33.7, minor difference (0.10) but update Fig. 3 if it’s supposed to be 33.8.

103: Is the similar time period, 1984-2020?, if so, add that it as “similar time period (1984-2020)” so its easy to follow this statement.

112: add sediment here, like “overlook sediment C accumulation”

113: add sediment/soil here like “sites of sediment/soil OC accumulation”

125-128: Great sentence!

131: please add the time frame and values so I can follow along with the statement. So annual erosion of lagoon OC for 1994-2017 is about 35 (?) of that orange bar and then comparing it to the backbarrier (blue bar) of 1984-2020, it is 33.7, so yes I can see that the annual erosion could negate the accumulation of the backbarrier if those numbers I listed are correct.

131-132. This sentence should add “assuming 100% of eroded OC is emitted to the atmosphere” because it can really only negate it if the accumulation is 100% buried and the eroded carbon is 100% emitted and the net value or outcome would be emitted, by about 1 Gg. Therefore, this sentence should say: “In fact, we find that the annual erosion of lagoon

OC (if 100% is emitted to the atmosphere) could negate the OC accumulated annually (assuming 100% is buried) in the entire backbarrier (Fig. 3)”

142: This is a great statement and I agree, we need more information about where eroded carbon goes and if it is remineralized immediately and gets emitted to the atmosphere.

143: add “net” to this sentence such as “as a net C source”

145: I think you need another sentence here that says by tracking the fate, the assumption of 100% loss of carbon from the ecosystem via erosion could be revisited and that OC erosion rates (which can help infer the flux of particulate carbon) may not equal to 100% emissions to the atmosphere and thus assessing what percent of eroded carbon goes to the atmosphere is important for carbon budgets.

187: add “matter” so it reads “organic matter”

190: Adding units for these terms in the equation would be helpful.

335: Figure 1 – Great figure!

340: add “sediment” here so it reads “stratigraphic section from sediments cores” to remind the reader these are sediment cores.

352: Figure 3. 1) Add “erosion” to the Y axis label to make it super clear this is the flux from erosion. It should read “Annual Rate of OC Erosion Flux”. 2) add soil to the legend for the blue color so it reads “average soil/sediment OC accumulation” 3) for the labels of 1984-2020, it should read “marsh soil”, “lagoon sediment” and “seagrass sediment”. These suggestions will help remind the reader that this is focused on erosion of soils and sediments.

352 Figure 3. I would use the term “gray bars” represent uncertainty instead of “grey lines” as at first it was difficult to find the grey lines.

Supplementary Information – suggestions by line number:

12 – change this to “Total Organic Carbon (TOC) versus Organic Matter Values and Uncertainties” – you should add the “Organic Matter” to this section header.

13 – LOI is a method and I find this confusing, what you measured is organic matter, so please use “Organic Matter (OM)(%) as the x axis label.

15 – I would change LOI to “organic matter” and leave the LOI in your Methods of the main manuscript. This LOI is a common technique and you already explained it in the Methods of your manuscript.

149 – Note that in Supplementary Table 5 there are superscript notations like *, but I don't see how they are defined in a footer or in the text.

Reviewer #1 (Remarks to the Author):

This manuscript calculates the loss of OC associated with erosion of the unvegetated environments within transgressing barrier islands. The manuscript characterizes a previously unaccounted for flux of OC from barrier islands into the coastal ocean associated with accelerating erosion and emphasizes further scrutiny of the role deep, unvegetated sediments play in coastal carbon dynamics. The provided supplementary materials and methods substantiate the conclusions and claims of the paper. I would appreciate a section specifically outlining how organic carbon was determined on the subset of sediments used to construct the conversion equation relating OC to LOI (see below). Though the manuscript relies heavily both on previously collected data and on a series of assumptions, the reasoning for each is clearly outlined and uncertainty is openly described throughout.

After careful scrutiny of the paper, I find it well-written and informative with no major flaws that would preclude publication. The manuscript is well-reasoned with a sound methodology and is a valuable addition to the blue carbon and barrier island erosion literature. Beyond that, it is an entertaining and insightful manuscript and I commend the authors on an excellent intellectual effort.

Thank you very much for your careful review and helpful comments. Please find below the details about our changes to the manuscript in response.

I have a few minor comments that are outlined below:

Lines 41 – 49. I understand why you're distinguishing between barrier islands and backbarrier marsh, but the two are not mutually exclusive. The backbarrier marsh is a component of the barrier island, which makes this paragraph a bit confusing. I'd suggest modifying the terminology to be more specific when you mention 'barrier island' to alleviate any confusion on the part of the reader.

We have modified the opening sentence (now lines 41–43) to more clearly identify the islands' beach and dune systems as the features that are doing the work—protecting, supplying sediment, transgressing, exposing—that we discuss in the body of the paragraph.

How was total organic carbon evaluated? I don't see any material about how these sediment subsamples were analyzed for TOC. Was inorganic carbon removed with acid? There seem to be quite a few points that seem anomalously high in TOC %, relative to LOI %. Could this be residual carbonate?

Thank you for pointing out this gap in our explanation of methods. We did, in fact, remove inorganic carbon with acid prior to analyzing samples for TOC. We have added this material into the Supplementary (lines 20–28):

“We analyzed a subset of sediment subsamples for total organic carbon (TOC) on a Costech Elemental Analyzer, model 4010, coupled to a ThermoFisher DeltaV Isotope Ratio Mass Spectrometer to determine conversion factors for organic matter (OM) to organic carbon (OC) (Supplementary Fig. 2). Prior to analysis, we freeze-dried and powdered samples and removed carbonates by adding 2 drops of 1N HCl and drying overnight at 60°C following methods outlined in refs. ^{1,2}. Average analytic precision (2- σ) for replicate measurements of marsh and lagoon TOC, respectively, were 0.06% and 0.18%. Uncertainties for all OC values were based on 95% confidence intervals and were propagated through to uncertainty estimates for final OC erosion rates”

The three anomalously high marsh TOC values are all from subsamples from dense marsh root mats within the top 40 cm of various cores. Thus, the high TOC values are likely a product of measuring younger and less decomposed marsh organic matter like roots and rhizomes. We analyzed 26 samples for TOC—from a range of depths within the marsh subunits of multiple islands—in order to gather a more representative OM to TOC relationships for marsh sediment.

Supplemental Data, Line 114: Can you define ‘surface marsh’? What is the depth on this?

We were using the term surficial to indicate < 1 meter, but we have removed the term “surface” and clarified depths by adding (new lines: 132–134):

“All cores collected as part of this study were opened, described, photographed, and sampled at a 10-cm resolution (for depths less than approximately 55 cm) in marsh peat and at 20–25 cm resolution through underlying lagoon deposits.”

Supplemental Data, Table 4: I’m a bit confused as to why the cores for Cedar Marsh are sporadically used. Why use this data for some things and not others?

We used all cores, including those collected in this study and those from the literature, in our general interpretation of stratigraphy. However, a common issue was that some cores did not penetrate the underlying Pleistocene sediments, and therefore could lead to underestimates of lagoon thickness. We did not use the cores we collected on Cedar Island to calculate lagoon thickness because they did not penetrate the underlying Pleistocene sediments. Instead, we used deeper cores from Cedar Island that are described in the literature² to estimate the thickness of lagoon sediments. On other islands for which published cores exist but do not reach Pleistocene sediments (that is, Assawoman and Smith Islands), we were forced to use a combination of cores collected as part of this study and from the literature, and to recognize that our estimates of lagoon thickness for these islands are conservative. Metompkin Island is an exception to this approach, as our one northern core, Met_G, does penetrate to Pleistocene sediments but through a relict tidal inlet that if used alone would greatly overestimate the thickness of the Holocene

lagoon unit. Finally, all cores collected as part of this study are used in averages of marsh and lagoon OC densities.

We clarified the rationale for these decisions in the Supplementary (new lines: 145–151) by re-writing this paragraph:

“With the exception of Met_G which reveals an anomalously thick Holocene unit (interpreted as a relict tidal inlet), cores collected from Assawoman, Metompkin, and Smith Islands as part of this study and others did not penetrate to the Pleistocene. Thus, we use all cores to inform stratigraphy and consider our lagoon thickness estimates for these islands to be conservative. This is in direct contrast to Cedar Island, where although cores collected as part of this study did not penetrate to the Pleistocene, we can approximate lagoon thickness from the two cores from the literature that did.”

Supplemental Data, Table 5: What are the notations (asterisks, other characters) for in the CAR column?

Thank you for catching this. We added these notations below the table.

While I understand that this is likely a mechanism of a slower publishing process, it is tough to evaluate the use of carbon accumulation rates (quite heavily) based on a study that is currently still in review at another journal and thus inaccessible. As a result, I can't speak to the appropriateness of using these measurements.

This paper was just published. DOI here: <https://doi.org/10.1007/s10021-023-00877-7>. We also want to note that although their C stock and accumulation values are reported in units of C, rather than OC, the majority (5 out of 6) of the sources used to calculate average marsh and seagrass stocks and rates reported units as OC (thus excluding inorganic C). However, if any C accumulation values incorporate inorganic C, our argument—that OC erosion along the beach/shoreface of migrating barrier islands outpaces OC accumulation in the backbarrier—is made stronger.

Throughout the manuscript, there are instances where “OC” and “C” appear to be used interchangeably. For example:

Thank you for pointing this out. For consistency and clarity, we converted most instances of “C” to “OC” except: a) where introducing general coastal C processes (for example, at the beginning of the Introduction and Discussion sections); b) where using terms like “blue carbon” which is widely used in reference to OC and c) where reporting on other literature that discuss findings related more generally to C rather than OC.

Line 194: “however, we account for lagoon sediment OC in our erosion terms, quantifying a maximum blue C loss term for erosion of the entire Holocene unit”.

To resolve the discrepancy between C and OC terminology, we have lengthened every instance of blue C to “blue carbon.”

Line 164: “blue C stored in wetland and thick lagoon sediments is largely ephemeral”.
What about inorganic carbon?

This is an interesting question. It is likely that fractions of previously buried inorganic C are re-exposed to erosion and potential dissolution during island migration/erosion, but this is not the focus of our work. Sediment along the beaches are a combination of quartz and some shell material¹. Thus, we expect inorganic C fluxes to comprise only a small fraction of total C fluxes along the beach- and shore-faces. Where we invoke “blue carbon” throughout the paper, we do mean organic C, as is consistent with the literature. Because this study does not attempt to quantify inorganic C we cannot speak to inorganic C fluxes, but we remain consistent with other blue carbon studies that are focused solely on organic, rather than inorganic, C.

To make this clear, we describe “blue carbon” ecosystems as those “recognized for [their] ability to store organic matter” (line 27–28).

Reviewer #2 (Remarks to the Author):

This paper describes a study of estimating sediment/soil organic carbon erosion fluxes of the Virginia Barrier Islands that have habitats of back barrier marshes, subtidal lagoon sediments and subtidal sediments with sea grasses. The researchers took sediment cores to estimate carbon content and how the erosion of these sediments and soils can help assess the annual rate of eroded organic carbon flux in this system. The main result in the study shows that the lagoon carbon erosion rate is higher than the soil/sediment carbon accumulation rate and thus suggesting that erosional processes can reduce the magnitude of net carbon sinks in these coastal landscapes. The authors argue that many studies focus on vegetated habitats but that these other subtidal sediment habitats are important to assess as carbon is likely eroding from them and could be influencing the net carbon budgets. This is a novel study because it includes non-vegetated habitats, and the methods are well described, and the results and discussion are well written. I provide many suggestions (by line numbers) to help the authors clearly explain their findings.

Thank you very much for your detailed comments and suggestions. Please find below the details about our changes to the manuscript in response.

Manuscript - suggestions with Line numbers:

11 – change “strength” to “magnitude”

Changed.

12: - add “soil” before “carbon” so it reads, “soil carbon”

Added.

18: - add “soil” before “carbon” so it reads, “soil carbon”

Added.

22: add “soil and sediment carbon” , so it reads “the impermanence of soil and sediment carbon” – I think this is important to remind the reader of where the buried or stored carbon is located.

Added.

24: Change “organic-rich sediment” to “organic-rich marsh soils and lagoon sediments” – to emphasize the 2 types of buried carbon this study evaluated.

Changed.

24: I also think “magnitude” is a better term than “strength” listed here.

Changed.

28: add “sediments and soils” after “bury C” so it reads, “bury C in sediments and soils” at rates...

Added.

29: Note you use “magnitude” here and I think this is the best term for your argument and study and I would suggest to only use this term in this paper and not “strength” to keep consistent and easy to read.

Thank you for the suggestion. Changed throughout.

31: add “soil” so it reads “soil C accumulation can result”

Added.

40: I don’t know what these terms “diminished and more fleeting” mean scientifically, I would simplify this and state “risk a blue C stock that is decreasing in time and space”

We have used “diminished” as an alternative to “decreasing in space” and “fleeting” as an alternative to “decreasing in time.” Nevertheless, we recognize that this is a stylistic preference, and defer to the editor for any suggestion.

50: add “net” so it reads “magnitude of the net C sink”...

Added.

50-51: The magnitude of the net C sink can also depend on other fluxes like the net methane flux. I would add the Chapin et al. 2006 citation to this sentence. Also note that the flux of net lateral transfer or flux of particulate carbon is a term commonly used in these carbon budgets and this term includes erosion. Thus, this study helps to estimate that flux of particulate carbon.

Chapin, F. S., G. M. Woodwell, J. T. Randerson, E. B. Rastetter, G. M. Lovett, D. D. Baldocchi, D. A. Clark, M. E. Harmon, D. S. Schimel, R. Valentini, C. Wirth, J. D. Aber, J. J. Cole, M. L. Goulden, J. W. Harden, M. Heimann, R. W. Howarth, P. A. Matson, A. D. McGuire, J. M. Melillo, H. A. Mooney, J. C. Neff, R. A. Houghton, M. L. Pace, M. G. Ryan, S. W. Running, O. E. Sala, W. H. Schlesinger, and E.-D. Schulze. 2006. Reconciling carbon-cycle concepts, terminology, and methods. *Ecosystems* 9:1041–1050.

And you can add that citation (by a superscript) to the word “balance” as this paper describes fluxes to measure for estimating the net ecosystem carbon balance.

Thank you very much for the paper recommendation and for the citation placement suggestion.

53: Agree with this statement that carbon budgets typically focus on the vegetated ecosystems. Great, thank you.

58: use the term “backbarrier marsh C soil accumulation rates” – marsh and soil need to be added here to be super clear to the reader.

Here, we used “backbarrier C accumulation rates” to include C accumulation in marsh, lagoon, and seagrass beds. To address your concern, we have modified the sentence to:

“Here, we combine geospatial data of barrier island retreat rates, organic carbon (OC) accumulation rates within backbarrier marsh soils and seagrass and lagoon sediments, and the OC content of eroding sedimentary facies to develop a regional-scale OC budget for the rapidly transgressing Virginia Atlantic coast (USA).”

63: Equation 1 was difficult to find, I suggest adding “equation” in front of the “(1)” on page 8 or line number 191.

Thank you, added.

96: edit “marsh exposure” to “long-term average marsh exposure rate” to be consistent with your figure.

Good suggestion, thank you.

97: Can you cite the figure or table that backs up this statement?

Yes, thank you for suggesting. We added these figures to the Supplementary information, as Supplementary Fig. 1.

101: In Figure 3 it lists 1994 and not 1997. So should this be 1994?

Yes, great catch.

102: add “soil/sediment” so it reads “annual soil/sediment OC accumulation rate”

Added.

102: It lists 33.8 and on Figure 3 it says 33.7, minor difference (0.10) but update Fig. 3 if it’s supposed to be 33.8.

Yes, again thank you for catching this.

103: Is the similar time period, 1984-2020?, if so, add that it as “similar time period (1984-2020)” so its easy to follow this statement.

Great, added.

112: add sediment here, like “overlook sediment C accumulation”

Added.

113: add sediment/soil here like “sites of sediment/soil OC accumulation”
Added.

125-128: Great sentence!

We appreciate it.

131: please add the time frame and values so I can follow along with the statement. So annual erosion of lagoon OC for 1994-2017 is about 35 (?) of that orange bar and then comparing it to the backbarrier (blue bar) of 1984-2020, it is 33.7, so yes I can see that the annual erosion could negate the accumulation of the backbarrier if those numbers I listed are correct.

This is a correct interpretation of our statement, and we have added the time frame and values to the sentence in question, so that it now reads:

“In fact, we find that just the average annual erosion of lagoon OC ($33.4 \text{ Gg} \pm 9.8 \text{ OC yr}^{-1}$; 1994–2017 C.E.) could negate the OC accumulated annually in the entire backbarrier averaged over a similar time period ($33.8 \pm 6.0 \text{ Gg OC yr}^{-1}$; 1984–2020 C.E.) (Fig. 3).”

Please note that in the revised manuscript, we modified the exclusion criteria for sediment subsamples (to exclude samples with incomplete data), and thus a few of our values shifted slightly.

131-132. This sentence should add “assuming 100% of eroded OC is emitted to the atmosphere” because it can really only negate it if the accumulation is 100% buried and the eroded carbon is 100% emitted and the net value or outcome would be emitted, by about 1 Gg. Therefore, this sentence should say: “In fact, we find that the annual erosion of lagoon OC (if 100% is emitted to the atmosphere) could negate the OC accumulated annually (assuming 100% is buried) in the entire backbarrier (Fig. 3)”

Thank you for the suggestion, and we agree. However, by adding the values and time periods as requested in the previous suggestion, adding more text to this sentence would make it too cumbersome. We have followed advice and now use the phrasing “could negate” in this sentence. However, we address the assumptions related to burial and mineralization in the next paragraph, where they can be treated more fully. Key sentences in that paragraph read:

“However, fully classifying the VBI chain as a net OC source would require tracking the fate of this shoreface-eroded OC, which may include remineralization, offshore burial, or possibly transport and redistribution to the backbarrier through tidal inlets. Nevertheless, the imbalance we measure between annual rates of backbarrier OC accumulation and shoreface OC erosion implies that, at the very least, barrier-island transgression results in a coastal OC sink that is far more tenuous than commonly assumed. “

142: This is a great statement and I agree, we need more information about where eroded carbon goes and if it is remineralized immediately and gets emitted to the atmosphere.

Thank you.

143: add “net” to this sentence such as “as a net C source”

Changed.

145: I think you need another sentence here that says by tracking the fate, the assumption of 100% loss of carbon from the ecosystem via erosion could be revisited and that OC erosion rates (which can help infer the flux of particulate carbon) may not equal to 100% emissions to the atmosphere and thus assessing what percent of eroded carbon goes to the atmosphere is important for carbon budgets.

Thank you for the suggestion. We have added the underline sentence to address the point that our comparisons are assuming partial or full remineralization of eroded OC, in line with C budgets from other studies:

“Including only marsh OC in these budgets would erroneously suggest that the VBI remains a strong sink for OC, netting an average 11.5 Gg OC yr⁻¹ over the past two decades (Fig. 3.) Like other landscape-scale carbon budgets^{6,12,20,23}, our work assumes that eroded carbon represents a source of carbon to the atmosphere or to non-coastal ecosystems.”

187: add “matter” so it reads “organic matter”

Added.

190: Adding units for these terms in the equation would be helpful.

We added these in the paragraph below the equation.

335: Figure 1 – Great figure!

Thank you! Glad you like it.

340: add “sediment” here so it reads “stratigraphic section from sediments cores” to remind the reader these are sediment cores.

Added.

352: Figure 3. 1) Add “erosion” to the Y axis label to make it super clear this is the flux from erosion. It should read “Annual Rate of OC Erosion Flux”. 2) add soil to the legend for the blue color so it reads “average soil/sediment OC accumulation” 3) for the labels of 1984-2020, it should read “marsh soil”, “lagoon sediment” and “seagrass sediment”. These suggestions will help remind the reader that this is focused on erosion of soils and sediments.

We decided to label the Y axis “OC flux” in order to keep the y-axis representative of all four bars, including the blue bar which shows accumulation rather than erosion. Thank you for suggestions 2) and 3) above which we have followed.

352 Figure 3. I would use the term “gray bars” represent uncertainty instead of “grey lines” as at first it was difficult to find the grey lines.

Agreed and added.

Supplementary Information – suggestions by line number:

12 – change this to “Total Organic Carbon (TOC) versus Organic Matter Values and Uncertainties” – you should add the “Organic Matter” to this section header.

Thank you, changed.

13 – LOI is a method and I find this confusing, what you measured is organic matter, so please use “Organic Matter (OM)(%) as the x axis label.

Yes, good call here.

15 – I would change LOI to “organic matter” and leave the LOI in your Methods of the main manuscript. This LOI is a common technique and you already explained it in the Methods of your manuscript.

Yes, good point, thank you.

149 – Note that in Supplementary Table 5 there are superscript notations like *, but I don’t see how they are defined in a footer or in the text.

Thank you for catching this. We added these notations below the table (which is now Table 4 to follow reference order in manuscript). Now it appears as:

Supplementary Table 4. Estimated Annual Carbon Accumulation Rates (CAR) for the Virginia Backbarrier Marshes, Seagrass Beds, and Lagoons.

Habitat	VBI Backbarrier Area* (km ²)	CAR [†] (g OC m ⁻² yr ⁻¹)	Total habitat CAR (Gg OC yr ⁻¹)	Average CAR between 1984 and 2020 (Gg OC yr ⁻¹)
Marsh	1984: 273.5 ± 14.2	78.4 ± 18.9*	1984: 21.4 ± 5.3	21.1 ± 5.2
	2020: 264.7 ± 9.7		2020: 20.8 ± 5.1	
Seagrass	1984: 0.0	40.1 ± 2.9*	1984: 0	0.6 ± 0.2
	2020: 29.3		2020: 1.2 ± 0.3	
Lagoon	1984: 473.8 ± 31.0	26.4 [§]	1984: 12.5 ± 3.2	12.1 ± 2.9
	2020: 445.4 ± 20.0		2020: 11.7 ± 2.5	
Total			1984: 33.9 ± 6.2	33.8 ± 6.0
			2020: 33.7 ± 5.7	

Note: These data feed into Fig. 3 and are used as a comparison to the average annual OC erosion rate of the VBI from 1994–2017 C.E.

*Values adapted from ref. ¹⁶ to extend from Assawoman in the north to Smith in the south, excluding all other VBI.

[†] Average CAR values include above- and belowground biomass and soil OC. CAR values from ref. ¹⁶ and uncertainties adapted from ref. ¹⁶.

[§]Combined average VBI lagoon %TOC from ref. ¹⁷ and average VBI lagoon sedimentation rates from ref. ¹⁸; no standard error reported.

References:

1. Finkelstein, K. & Ferland, M. A. Back-barrier response to sea-level rise, Eastern Shore of Virginia. *The Society of Economic Paleontologists and Mineralogists* (1987).
2. Shawler, J. L., Ciarletta, D. J., Lorenzo-Trueba, J. & Hein, C. J. Drowned foredune ridges as evidence of pre-historical barrier-island state changes between migration and progradation. *Coastal Sediments Proceedings* 158–171 (2019).